# A pilot study of the knowledge, awareness and perception of prostate cancer in Ghanaian women

**Ebenezer Wiafe**[1,2]*, **Kofi Boamah Mensah**[1,3], **Frasia Oosthuizen**[1], **Varsha Bangalee**[1]

**1** Discipline of Pharmaceutical Sciences, College of Health Sciences, University of KwaZulu-Natal, Durban, South Africa, **2** Clinical Pharmacy Services Unit, Directorate of Pharmacy, Ho Teaching Hospital, Ho, Ghana, **3** Department of Pharmacy Practice, Faculty of Pharmacy and Pharmaceutical Sciences, College of Health Sciences, Kwame Nkrumah University of Science and Technology, Kumasi, Ghana

* weben38@gmail.com

## Abstract

### Background

The African prostate cancer epidemiological trend has reported the late detection of the disease and resultant high mortality rate. Considering the economic position of the African continent, which often contributes to high mortality, it has become imperative to investigate cost-effective means of improving the timely detection of prostate cancer. This study, the third developmental phase of a robust Akan tool, aimed at conducting an external pilot survey to investigate the practicability of the tool in studying prostate cancer awareness in women.

### Method

The study was conducted in one of the biggest markets in Ghana employing a quantitative approach and recruiting 400 females from the age of 18 years. Post-ethical approval and study subjects' consent, the participants randomly responded to the Akan tool and the data was electronically entered in the presence of the participants. The data, entered in the Microsoft Excel spreadsheet, were analysed with the SPSS software (version 25). The results were presented as frequencies and percentages, with an assessment of the tool's reliability.

### Results

A Cronbach's alpha reliability coefficient of 0.9030 was calculated. The majority (83.50%) of the participants belonged to the Akan tribe and were fluent in the Akan language. None of the knowledge items on the signs and symptoms, and risk factors of the disease had correct responses from more than 25.00% and 20.00% of the participants respectively whilst knowledge items on the causes of prostate cancer received varied responses. The participants were aware of the disease and had a positive perception.

**Data Availability Statement:** All relevant data are within the manuscript.

**Funding:** The study was supported by the 2021 early career research grant award by the Royal

Society of Tropical Medicine and Hygiene in collaboration with the National Institute of Health Research.

**Competing interests:** The authors have declared that no competing interests exist.

## Conclusions

The pilot survey adequately tested the Akan tool and suggested various modifications to the tool and the study methodology. The tool exhibited acceptable reliability and could be applied to targeted populations to investigate the awareness of prostate cancer in women.

## Background

According to a study by Rashid et al. (2007), women reported poor prostate cancer knowledge and their inability to contribute to the timely detection of prostate cancer and stressed the need for education and inclusiveness [1]. Drawing on the evidence from a systematic review conducted by Wiafe and colleagues (2021), it has become relevant to investigate the knowledge, awareness and perception of prostate cancer (PCa) in African women [2]. According to the authors, none of the seven primary (7) studies that were included in their review was conducted in Africa. The review also reported that 5 out of the 7 studies recruited participants of African descent. Some African literature on PCa knowledge, awareness, perception and practices of recruited participants mainly focused on males, ignoring the female gender [3–5]. Therefore, these scholarly gaps concerning this subject area require further research.

The epidemiological pattern of PCa has been studied by many researchers and international agencies. There is evidence suggesting that the developed world has a higher PCa incidence compared to developing countries [6]. This observation has been linked to the robust screening programs that exist in the developed world, of which the United States of America has been captured. The developed countries have reported lower prevalence and mortality rates compared to the developing countries. These positive outcomes are evidence of the institution of early detection policies and advanced management protocols [6, 7]. The higher prevalence rates that have been reported from countries of lower economic classification have mainly been attributed to high mortality rates. The Ghanaian data has not deviated from the reports of international authorities including the Global Cancer Incidence, Mortality and Prevalence (GLOBOCAN) [8–10]. This phenomenon has warranted the need for the implementation of affordable approaches such as the administration of educational interventions to improve the early detection of PCa in resource-constraint countries [11].

The development and application of educational interventions require the availability of robust knowledge assessment tools. With the high success rate of educational interventions [11, 12] and the lack of Ghanaian PCa knowledge assessment tools [2], it has become imperative to develop and pilot such a tool. Through the efforts of a team of researchers, an Akan PCa awareness tool has been developed, validated, and adopted in this pilot study [13]. This external pilot study aimed to determine the reliability of the Akan tool in studying PCa awareness in a standard sample of women and to evaluate the practicability of the proposed methodological approaches in the application of this already validated Akan tool in PCa awareness assessment.

## Methodology

The research protocol received a local ethical authorisation from the Committee on Human Research, Publications and Ethics (CHRPE) of the Kwame Nkrumah University of Science and Technology (KNUST)–Approval reference: CHRPE/AP/110/21. The approval from the CHRPE was submitted to the Biomedical Research Ethics Committee (BREC) of the

University of KwaZulu-Natal (UKZN) for final authorisation (Approval reference: BREC/00002740/2021) to commence the study. The researchers were guided by the principles of cross-sectional quantitative research design utilizing an already developed and validated Akan tool [13]. The Akan tool had undergone face and content validation, and reliability assessment exhibiting acceptable results which warranted its application in this pilot study. According to the requirements of the ethics committees, the participants appended signatures to written informed consent forms before participation.

## Study settings and subjects

The study was conducted on the Kejetia Market (the Kumasi Central Market) in the Ashanti region of Ghana where the ground floor of the market was excluded. This exclusion was because the potential participants on the ground floor had been engaged in the development of the Akan tool, the second stage of the tool development process. According to the principles of random sampling, the remaining floors of the market were divided into four (4) parts. The researchers randomly selected one part and did a subdivision into four (4) parts. The principal investigator (PI) randomly selected and collected data from 100 participants per subdivision. The study considered only females (18 years and above) for inclusion.

## Sample size justification

With an estimated adult female population of 25,000 at the study site, a sample of 400 adult females was recruited for the study. Considering a distribution response rate (p) of 50%; a precision level (e) of 5%; a confidence interval (CI) of 95%; a standard normal variance (Z-score) of 1.96 to obtain a power of 95% confidence interval and a type 1 error probability of 5%; the minimum sample size (n) according to the Cochran's formula (where; 1- p = proportion of the non-response distribution rate) is 384. According to a published table by Israel (1992), employing Yamane's formula, a minimum sample size of 394 is appropriate [14]. Hence, to accommodate for a non-response, to ensure a stronger statistical power and effect size, and to satisfy the requirement of the initial objective of the study, the sample of 400 adult females is justified.

## Data collection

The collection of the data was done solely by the Principal Investigator (PI) through electronic entry using google forms, according to the random sampling technique. The PI approached the respondents, obtained informed consent, read the content of the printed tool to the respondents, and completed the entries electronically in the presence of the respondents. The PI solely collected the data to inform him of the possible challenges of the data collection technique and to further equip him to propose changes to the tool and the data collection approach and train the required number of data collectors for the main study.

## Data management and statistical analysis

The captured data was exported onto a Microsoft Excel spreadsheet, cleaned, and checked for errors to ensure data completeness. The demographic features of the participants were regrouped during the data management process. Notably: age and ethnicity were reclassified into 2 groups; and marital status, educational level, and religious affiliation were re-categorized into 3 groups to aid data analysis. The reclassification was in response to responses being too little for data analysis in the original groupings of the listed demographics.

The knowledge of women on PCa was measured using 21 questionnaire items on the signs and symptoms, causes, and risk factors. Assessing the awareness of the participants about PCa involved 8 questionnaire items of which 4 had "yes", "no" or "don't know" responses. The knowledge and awareness questions were scored on a 2-point Likert scale of correct response and wrong response. The scale scored 1 for correct response and 0 for wrong response including the option of "don't know". The participants' perception about PCa was assessed with 5 questionnaire items each for attitude and belief domains. The 10 questionnaire items on perception were scored on a 2-point Likert-like scale of "agree" and "disagree". The scale was scored as agree (1) and disagree (0) for the positive items, and disagree (1) and agree (0) for the negative items.

The polished data was analysed using SPSS version 25. The results for continuous variables were presented as mean ± standard deviation. Categorical variables were presented as frequencies (n) and percentages (%). The reliability of the validated tool [13] employed in this external pilot study was reassessed using Cronbach's alpha. The corresponding author is the original copyright holder of the validated tool employed in this study.

## Results

The redetermination of the reliability coefficient for the utilized validated tool was 0.9030 and lay between the previously reported range of 0.7808 and 0.9209 [13]. As observed in Table 1,

**Table 1. Demographical features of the study subjects (n = 400).**

| Variables | n (%) |
|---|---|
| **Age** (Mean ± SD) | 45.02 ± 10.66 |
| **Age** (Minimum) | 18 |
| **Age** (Maximum) | 73 |
| **Age** (Ranges) | |
| 15–29 | 36 (9.00) |
| 30–49 | 213 (53.25) |
| 50+ | 151 (37.75) |
| **Marital Status** | |
| Never married | 36 (9.00) |
| Ever married | 364 (91.00) |
| **Highest Educational level** | |
| Primary | 201 (50.25) |
| Higher education | 118 (29.50) |
| No education | 81 (20.25) |
| **Religious Affiliation** | |
| Christianity | 367 (91.75) |
| Islamic | 32 (8.00) |
| Other | 1 (0.25) |
| **Ethnic Background** | |
| Akan | 334 (83.50) |
| Other ethnicity | 66 (16.50) |
| **Market Association** | |
| Clothes Sellers | 20 (5.01) |
| Shoe Sellers | 11 (2.76) |
| Kejetia Market Union | 1 (0.25) |
| None | 368 (91.98) |

the mean age of the participants was 45.02 ± 10.66 years with the highest proportion within the age bracket of 30–49 years (53.25%). About a ninth of the participants were ever married. Educational level was in the order of primary level (50.25%), higher education (29.5%), and no education (20.25%). The study sample was dominated by Christians (91.75%) and Akans (83.50%), and the majority of them did not belong to any market associations.

Table 2 indicates the responses of the study subjects on their knowledge of the signs and symptoms, causes, and risk factors of PCa. Amongst the nine (9) items that assessed the participant's knowledge about the signs and symptoms of PCa, none of these knowledge items received a correct response from at least 25.00% of the respondents. Between 61.00–95.00% of the respondents did not know the response to give regarding these knowledge items. In the assessment of knowledge about the causes of PCa, five (5) knowledge questions were administered. Amongst these, 83.50% of the respondents correctly indicated that curses could not cause prostate cancer. Approximately, one-fourth of the participants correctly responded that some pharmaceutical agents could cause PCa, whilst 27.25% of the study subjects wrongly implicated mobile phones as a cause of PCa. When the participants were asked about the possibility of infectious diseases causing PCa, 80.25% could not choose between "agree" or "disagree". About 70% of the participants could not also tell if PCa had potential causes. The third domain of knowledge assessment was examined with seven (7) items of which none had a 20.00% correct response from the participants. As low as 4.00% and approximately 10.00% of

**Table 2. Knowledge about prostate cancer (n = 400).**

| Variables | Agree n (%) | Disagree n (%) | Don't know n (%) |
|---|---|---|---|
| **Knowledge on signs and symptoms of prostate cancer** | | | |
| Poor urine flow. **TRUE** | 22 (5.50) | 0 (0.00) | 378 (94.50) |
| Frequent urination at night which disturbs sleep. **TRUE** | 31 (7.75) | 1 (0.25) | 368 (92.00) |
| Sexual weakness or impotence. **FALSE** | 49 (12.25) | 21 (5.25) | 330 (82.5) |
| Waist pain. **TRUE** | 28 (7.00) | 1 (0.25) | 371 (92.75) |
| Blood in urine. **TRUE** | 80 (20.00) | 8 (2.00) | 312 (78.00) |
| Abrupt inability to urinate. **TRUE** | 21 (5.25) | 1 (0.25) | 378 (94.50) |
| Urgent need to pass urine. **TRUE** | 22 (5.50) | 1 (0.25) | 377 (94.25) |
| Abdominal pains. **TRUE** | 20 (5.00) | 0 (0.00) | 380 (95.00) |
| There are no early signs and symptoms of prostate cancer. **TRUE** | 96 (24.00) | 60 (15.00) | 244 (61.00) |
| **Knowledge on the causes of prostate cancer** | | | |
| Infectious diseases. **FALSE** | 62 (15.50) | 17 (4.25) | 321 (80.25) |
| Putting mobile phones in your pocket. **FALSE** | 109 (27.25) | 157 (39.25) | 134 (33.50) |
| Curses. **FALSE** | 21 (5.25) | 334 (83.50) | 45 (11.25) |
| Some drugs for the treatment of diseases. **TRUE** | 101 (25.25) | 46 (11.50) | 253 (63.25) |
| The cause of prostate cancer is unknown. **TRUE** | 84 (21.00) | 39 (9.75) | 277 (69.25) |
| **Knowledge on the risk factors of prostate cancer** | | | |
| Increasing age. **TRUE** | 16 (4.00) | 11 (2.75) | 373 (93.25) |
| Excessive alcohol consumption. **TRUE** | 52 (13.00) | 30 (7.50) | 318 (79.50) |
| Excessive smoking of cigarette. **TRUE** | 56 (14.00) | 24 (6.00) | 320 (80.00) |
| Being of Black decent. **TRUE** | 78 (19.50) | 72 (18.00) | 250 (62.50) |
| Family history of the disease. **TRUE** | 38 (9.50) | 222 (55.50) | 140 (35.00) |
| Vasectomy. **FALSE** | 137 (34.25) | 75 (18.75) | 188 (47.00) |
| Regular sexual intercourse. **FALSE** | 31 (7.75) | 27 (6.75) | 342 (85.50) |

Low Knowledge Level: 38.2% of participants; High Knowledge Level: 61.8% of participants

the respondents respectively correctly attributed increased age and family history as risk factors of PCa, and African decency received the highest correct response (19.25%).

The awareness about PCa was studied with eight (8) items of which seven (7) applied to all the participants. Amongst the participants, 396 (99.00%) had heard of PCa from sources such as the media only (364), friends only (1), family members only (1), healthcare providers only (1), and other multiple sources (29). Table 3a indicates the awareness responses of the participants where 77.75% of the respondents affirmed none of their family relatives had suffered PCa. About 20.00% of the respondents were aware of the fact that PCa had claimed the lives of many men and was a malignant medical condition. Over a third of the study subjects indicated that PCa was a disease of only men whilst 58 participants (14.50%) responded that the disease affects both sexes. When the participants were asked about the possible transmission of PCa through sexual intercourse, "disagree" and "don't know" respectively had 48.50% and 51.50%

**Table 3. Awareness about prostate cancer (n = 400).**

**a**

| Variable | Yes n (%) | No n (%) | Don't know n (%) |
|---|---|---|---|
| Has any of your relatives been diagnosed with prostate cancer?* | 84 (21.00) | 311 (77.75) | 5 (1.25) |
| Prostate cancer kills more men than any other cancer found in men. **Correct** | 74 (18.50) | 21 (5.25) | 305 (76.25) |
| Prostate cancer can spread to other parts of the body. **Correct** | 82 (20.50) | 23 (5.75) | 295 (73.75) |
| Men can transfer prostate cancer to women through sexual intercourse. **Wrong** | 0 (0.00) | 194 (48.50) | 206 (51.50) |
| Which people does prostate cancer affect? | | | |
| Only men. **Correct** | 333 (83.25) | | |
| Only women. | 1 (0.25) | | |
| Both. | 58 (14.50) | | |
| Don't know. | 8 (2.00) | | |

**b**

| Where can prostate cancer be found? | n (%) |
|---|---|
| Penis. | 248 (62.00) |
| Lower abdomen. | 53 (13.25) |
| Abdomen. | 47 (11.75) |
| Breast. | 5 (1.25) |
| Testicles. | 2 (0.50) |
| Prostate gland. **TRUE** | 2 (0.50) |
| Stomach. | 2 (0.50) |
| Bladder. | 2 (0.50) |
| Lungs. | 1 (0.25) |
| All parts of the body. | 5 (1.25) |
| Penis and other parts of the body. | 6 (1.50) |
| Lower abdomen and other parts of the body. | 4 (1.00) |
| Bladder and other parts of the body. | 1 (0.25) |
| Breast and other parts of the body. | 1 (0.25) |
| Don't know. | 21 (5.25) |

*No wrong or correct response

responses. As indicated in Table 3b, only two (2) participants knew PCa affects the prostate gland whilst over 60.00% of the participants (248) wrongly tagged the penis as the organ that suffers PCa.

The perception of women about PCa was studied with five (5) items each for attitude and beliefs, Table 4. All the attitude items received positive responses in the range of 84.50–100.00% of participants. These positive responses reflected good attitudes except the response which indicated that 84.50% of the respondents would deny their husbands' diagnosis of PCa sexual satisfaction. Table 4 revealed that all the participants found PCa education as an indispensable intervention, and the majority of the participants placed importance on PCa screening. On the belief arm, whilst 87.75% of the participants believed every man could develop PCa, a higher proportion of the participants disagreed that PCa is a spiritual disease (90.25%) nor a family disease (66.00%). Almost all the participants agreed PCa could be cured when diagnosed early. A third of the participants believed nothing could be done to save the life of a PCa patient.

## Discussion

This pilot study served as the third stage of the development of a robust Akan tool, with appropriate psychometric properties, aimed at studying prostate cancer awareness amongst Ghanaian women. The first stage of the tool development involved the conduct of a systematic review which brought together various questionnaire items [2]. The questionnaire items were then combined to develop an English version data extraction tool which was translated into the Akan language, certified, and subjected to validity and reliability assessments as part of the second stage of tool development [13]. To further stress the reliability of the Akan tool, in this pilot study, the researchers calculated Cronbach's alpha reliability coefficient as 0.9030. Previously, the Cronbach's alpha for the various sections of the Akan tool had ranged between 0.7808 and 0.9209 and was considered to exhibit sound reliability. The reported reliability coefficient falls within this reported range and hence, the pilot study has been successful in strengthening the reliability of the Akan tool.

With the administration of the Akan tool to a standard sample of participants, we are justified to report the findings of this external pilot study as tables [15]. The study adhered to the

**Table 4. Perception about prostate cancer (n = 400).**

| Variable | Agree n (%) | Disagree n (%) |
|---|---|---|
| **Attitude towards prostate cancer** | | |
| A woman can take her husband/relative/friend to be screened for prostate cancer. | 388 (97.00) | 12 (3.00) |
| A woman can recommend to men to screen for prostate cancer. | 396 (99.00) | 4 (1.00) |
| Prostate cancer screening must be free for every man. | 352 (88.00) | 48 (12.00) |
| Prostate cancer education is important. | 400 (100.00) | 0 (0.00) |
| Men diagnosed with prostate cancer must not have sex with their wives. | 338 (84.50) | 62 (15.50) |
| **Beliefs about prostate cancer** | | |
| Prostate cancer is a family disease. | 136 (34.00) | 264 (66.00) |
| Any man can develop prostate cancer. | 351 (87.75) | 49 (12.25) |
| Prostate cancer is a spiritual disease. | 39 (9.75) | 361 (90.25) |
| Prostate cancer can be cured when diagnosed early. | 399 (99.75) | 1 (0.25) |
| Nothing can be done to save a person with prostate cancer. | 301 (75.25) | 99 (24.75) |

Negative Perception: 32.9% of participants; Positive Perception: 67.1% of participants

proposed methodology and successfully collected data to prove the feasibility of the tool for studying PCa awareness. As reported in Table 1, the ethnic distribution of the study subjects supported the relevance of the Akan tool because 83.50% of the participants were Akans. Also, during the data collection process, none of the participants was found to not have been fluent in the Akan language. The application of the tool did not warrant the need for translators, and this might have boosted participants' comprehension [16], accounted for a complete response rate [17], and the preservation of the reliability coefficient [18]. The findings of the study resulted in the preposition of various modifications to the tool and the methodological procedure. Hence, the pilot study has successfully proven the applicability of the Akan tool to study PCa awareness in the target population.

## Recommended modifications to study methodology and tool

Relying on the evidence from the study, the research team makes the following non-exhaustive recommendations to modify the tool, and guide its application:

1. To collect data using printed questionnaires to eliminate potential participants' fear of being audio-visually recorded.

2. To make the developed tool applicable to any Akan-speaking community, the researchers propose the elimination of the last demographic item.

3. To ensure the even scoring of the awareness items, we propose the rewording, the reduction of awareness items, and the restriction of response keys to "yes" and "no".

4. Due to the absence of an Akan word for the prostate gland, the researchers propose the modification of the 11[th] awareness item to prevent participants from providing wrong responses.

5. To eliminate the option of "don't know" as a response to knowledge and awareness assessment items. This recommendation is justified by the large responses that were received by "don't know" which made it impossible to determine the firm position of respondents concerning knowledge and awareness items. This modification would reduce the total time participants would spend responding to the tool.

## Strengths and limitations

The study was strengthened through the following approaches;

1. The elimination of selection bias through the random sampling of the study subjects.

2. The recruitment of a standard sample size in this pilot study improved the acceptability of the study findings.

3. The use of a validated and reliability-tested tool aimed to improve the quality of the study.

4. The application of a locally developed tool enhanced the comprehension of the participants and improved the quality of responses. This also prevented the possible misinterpretation of the questionnaire items by the study subjects.

The study was faced with the following limitations;

1. The data collection procedure, which involved the electronic entry of data in the presence of the participants, made some potential participants reject inclusion for the fear of possible audio-visual recording.

2. The Akan language does not have a word for the prostate gland. This affected responses that were provided to the location of prostate cancer.

## Conclusion

In conclusion, this pilot study has served its purpose by affirming the reliability of the Akan tool, adequately testing the feasibility of the tool to study PCa awareness amongst Ghanaian women, and proposing various changes to the study design and the Akan tool. Most importantly, the researchers recommend the implementation of these methodological and tool modifications, and a further subjection of the modified tool to a psychometric analysis in a different population of similar sociodemographic parameters. Given this, we would implement these recommendations and recruit participants from the New Agogo Community Market in the Ashanti region of Ghana in an attempt to improve the robustness of the Akan tool through a psychometric study.

## Acknowledgments

The research team gives recognition to Prof. Charles Marfo.

## Author Contributions

**Conceptualization:** Ebenezer Wiafe, Kofi Boamah Mensah.

**Data curation:** Ebenezer Wiafe.

**Formal analysis:** Ebenezer Wiafe.

**Methodology:** Ebenezer Wiafe, Kofi Boamah Mensah, Frasia Oosthuizen, Varsha Bangalee.

**Project administration:** Ebenezer Wiafe.

**Supervision:** Kofi Boamah Mensah, Frasia Oosthuizen, Varsha Bangalee.

**Writing – original draft:** Ebenezer Wiafe.

**Writing – review & editing:** Kofi Boamah Mensah, Frasia Oosthuizen, Varsha Bangalee.

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
