## [Editor Report · Decision Letter 0]

23 Sep 2021

PONE-D-21-29398

A Pilot Study of the Knowledge, Awareness and Perception of Prostate Cancer in Ghanaian Women

PLOS ONE

Dear Dr. Wiafe,

Thank you for submitting your manuscript to PLOS ONE. After careful consideration, we have decided that your manuscript does not meet our criteria for publication and must therefore be rejected.

Specifically:

not using validated tools is the main reason for the rejection of the manuscript.

I am sorry that we cannot be more positive on this occasion, but hope that you appreciate the reasons for this decision.

Yours sincerely,

Forough Mortazavi

Academic Editor

PLOS ONE

Additional Editor Comments:

Dear authors,

Thank you for working on prostate cancer. Thank you for conducting research on prostate cancer. In your cross-sectional study, knowledge, awareness and perception of prostate cancer in 400 Ghanaian women were investigated using the 'Akan' tool. You state that this study is the third developmental phase of the robust Akan tool. It seems that the scale has not been validated systematically yet because there is no mention any previous studies on the validity of the scale in your reported references. In validation studies, the content validity as well as the construct validity of a scale are investigated. If there exist such validity studies with regard to 'Akan' they could shed light on the practicability of the scale in studying prostate cancer awareness. It is not therefore certain that this study used a valid instrument to investigate knowledge, awareness and perception of prostate cancer in Ghanaian women. This can affect the results and the conclusions.
---

## [Author Response · Author response to Decision Letter 0]

1 Oct 2021

ACADEMIC EDITOR’S COMMENT

Thank you for submitting your manuscript to PLOS ONE. After careful consideration, we have decided that your manuscript does not meet our criteria for publication and must therefore be rejected.

Specifically: not using validated tools is the main reason for the rejection of the manuscript.

RESPONSE

Please a validated tool was used in this pilot study.

ADDITIONAL EDITOR COMMENTS:

You state that this study is the third developmental phase of the robust Akan tool. It seems that the scale has not been validated systematically yet because there is no mention any previous studies on the validity of the scale in your reported references. In validation studies, the content validity as well as the construct validity of a scale are investigated. If there exist such validity studies with regard to 'Akan' they could shed light on the practicability of the scale in studying prostate cancer awareness. It is not therefore certain that this study used a valid instrument to investigate knowledge, awareness and perception of prostate cancer in Ghanaian women. This can affect the results and the conclusions.

RESPONSE

The study is the third developmental phase because;

1. The first phase involved a systematic review which discovered various studies concerning the subject. Kindly find the link: https://doi.org/10.1186/s13643-021-01695-5

2. The second phase brought together the tools that were used in the studies that were discovered in 1 above. These tools were then used to develop a single English version tool which was translated into the Akan language, with certification. The Akan version then went through a validation and reliability analysis. This second phase represents the previous study referred to in your comment. Kindly find the link to the validity study: http://waocp.com/journal/index.php/apjcc/article/view/644

Please a validated tool was used in the study. The presentation of this pilot study as the third phase is not intended to present this study as a study of low quality because the authors even used a standard sample size. Also, the purpose of pilot studies is for testing (for example, tools), make recommendations, and not present strong scientific findings.

The robust Akan tool is the tool the authors intend to present to the scientific community in the fourth phase of the project.

Thank you.

---

## [Decision Letter · Decision Letter 1]

18 Apr 2022

A Pilot Study of the Knowledge, Awareness and Perception of Prostate Cancer in Ghanaian Women

PONE-D-21-29398R1

Dear,

We’re pleased to inform you that your manuscript has been judged scientifically suitable for publication and will be formally accepted for publication once it meets all outstanding technical requirements.

Kind regards,

Muhammad Shahzad Aslam, Ph.D.,M.Phil., Pharm-D

Academic Editor

PLOS ONE

Reviewers' comments:

Reviewer's Responses to Questions

**Comments to the Author**

1. If the authors have adequately addressed your comments raised in a previous round of review and you feel that this manuscript is now acceptable for publication, you may indicate that here to bypass the “Comments to the Author” section, enter your conflict of interest statement in the “Confidential to Editor” section, and submit your "Accept" recommendation.

Reviewer #1: All comments have been addressed

Reviewer #2: (No Response)

2. Is the manuscript technically sound, and do the data support the conclusions?

Reviewer #1: Yes

Reviewer #2: Yes

3. Has the statistical analysis been performed appropriately and rigorously? 

Reviewer #1: Yes

Reviewer #2: Yes

4. Have the authors made all data underlying the findings in their manuscript fully available?

Reviewer #1: Yes

Reviewer #2: Yes

5. Is the manuscript presented in an intelligible fashion and written in standard English?

Reviewer #1: Yes

Reviewer #2: Yes

6. Review Comments to the Author

Reviewer #1: The main concern of the academic editors for this manuscript was the validation of the tool. I have searched wisely and by taking the links from the author's response page, I found that the tool is already validated. 

The previous study result on the validation of the same tool showed that a forty-five (45) member Akan questionnaire was successfully developed and certified. The average scores for all parameters employed in the face validation were greater than 4. The content validity index was within the range of 0.90–0.99, while the Cronbach’s alpha for both test periods was within the range of 0.7808–0.9209.

Finally, they conclude that the Akan questionnaire had acceptable validity and reliability outcomes. Therefore, the questionnaire was considered appropriate for assessing the knowledge, awareness, and perception of Ghanaian women about prostate cancer.

Therefore, the tool is validated, and I think the manuscript is eligible for this phase.

In addition, other points raised are well addressed by the authors, such as competing interests and financial issues.

(HINT: I strongly focused on previous comments raised by the editor). 

I hope this manuscript will be accepted by the editor and we will also comment on the body of the manuscript.

Thank you.

Reviewer #2: If possible, please use some more recent papers as more than half of the references are more than 5 years old.

7. PLOS authors have the option to publish the peer review history of their article (what does this mean?). If published, this will include your full peer review and any attached files.

Reviewer #1: No

Reviewer #2: **Yes: **

---

## [Editor Report · Acceptance letter]

21 Apr 2022

PONE-D-21-29398R1 

A pilot study of the knowledge, awareness and perception of prostate cancer in Ghanaian women 

Dear Dr. Wiafe:

I'm pleased to inform you that your manuscript has been deemed suitable for publication in PLOS ONE. Congratulations! Your manuscript is now with our production department. 

Kind regards, 

on behalf of

Dr. Muhammad Shahzad Aslam 

Academic Editor

PLOS ONE